# SNX-PXA-RGS-PXC Subfamily of SNXs in the Regulation of Receptor-Mediated Signaling and Membrane Trafficking

**DOI:** 10.3390/ijms22052319

**Published:** 2021-02-26

**Authors:** Bibhas Amatya, Hewang Lee, Laureano D. Asico, Prasad Konkalmatt, Ines Armando, Robin A. Felder, Pedro A. Jose

**Affiliations:** 1The George Washington University, Washington, DC 20052, USA; bibhasamatya78@gwmail.gwu.edu; 2Department of Medicine, The George Washington University School of Medicine & Health Sciences, Washington, DC 20052, USA; lih@gwu.edu (H.L.); lasico@email.gwu.edu (L.D.A.); prk@email.gwu.edu (P.K.); iarmando@email.gwu.edu (I.A.); 3Department of Pathology, University of Virginia Health Sciences Center, Charlottesville, VA 22908, USA; raf7k@virginia.edu; 4Department of Pharmacology/Physiology, The George Washington University School of Medicine & Health Sciences, Washington, DC 20052, USA

**Keywords:** endosome, GPCR, receptor, RGS, signaling, SNX, trafficking

## Abstract

The SNX-PXA-RGS-PXC subfamily of sorting nexins (SNXs) belongs to the superfamily of SNX proteins. SNXs are characterized by the presence of a common phox-homology (PX) domain, along with other functional domains that play versatile roles in cellular signaling and membrane trafficking. In addition to the PX domain, the SNX-PXA-RGS-PXC subfamily, except for SNX19, contains a unique RGS (regulators of G protein signaling) domain that serves as GTPase activating proteins (GAPs), which accelerates GTP hydrolysis on the G protein α subunit, resulting in termination of G protein-coupled receptor (GPCR) signaling. Moreover, the PX domain selectively interacts with phosphatidylinositol-3-phosphate and other phosphoinositides found in endosomal membranes, while also associating with various intracellular proteins. Although SNX19 lacks an RGS domain, all members of the SNX-PXA-RGS-PXC subfamily serve as dual regulators of receptor cargo signaling and endosomal trafficking. This review discusses the known and proposed functions of the SNX-PXA-RGS-PXC subfamily and how it participates in receptor signaling (both GPCR and non-GPCR) and endosomal-based membrane trafficking. Furthermore, we discuss the difference of this subfamily of SNXs from other subfamilies, such as SNX-BAR nexins (Bin-Amphiphysin-Rvs) that are associated with retromer or other retrieval complexes for the regulation of receptor signaling and membrane trafficking. Emerging evidence has shown that the dysregulation and malfunction of this subfamily of sorting nexins lead to various pathophysiological processes and disorders, including hypertension.

## 1. Introduction

Receptor-mediated signaling and membrane trafficking processes are intimately interconnected with the endosomes [1]. Internalized receptors, including G protein-coupled receptors (GPCRs) and non-GPCRs, are sorted at endosomes, from which receptors are either delivered to the lysosome for degradation, recycled back to the plasma membrane, or delivered to the trans-Golgi network (TGN) and other organelles by receptor-specific pathways [2,3]. Sorting nexins (SNXs) play critical roles in these processes [4].

The SNX family has a phox homology (PX) domain, capable of phosphoinositide binding, which enables SNX targeting to endosomal membranes by binding to phosphatidylinositols, most commonly phosphatidylinositol 3-phosphate (PI(3)P) [5,6]. SNXs are widely expressed from yeast to mammals, whose PX domain, first identified in two subunits of the NADPH oxidase, p40^phox^ and p47^phox^, actively engages in protein–lipid and protein–protein interactions [5,6]. To date, 10 yeast and 33 mammalian SNXs have been identified [5,6,7,8]. Based on their domain architectures, the mammalian SNXs are divided into five subfamilies: SNX-PXA-RGS-PXC, SNX-FERM (protein 4.1/ezrin/radixin/moesin), SNX-BAR (Bin/Amphiphysin/Rvs), SNX-PX, and the unclassified SNX subfamilies [7,8] (Table 1).

## 2. SNX-PXA-RGS-PXC Subfamily Domain Structure and Biochemical Properties

The SNX-PXA-RGS-PXC subfamily is comprised of SNX13 (also known as RGS-PX1), SNX14, SNX19, and SNX25. This subfamily of SNXs contains two N-terminal helical transmembrane domains, followed by a PX-associated domain (PXA), a regulators of G protein signaling (RGS) domain, the PX domain, and a C-terminal PX-associated (PXC) domain [6,8]. 

Integrated transmembrane domains (IMDs), which are two, close short hydrophobic sequences, are involved in membrane tethering [4,6]. RGS domain, a unique domain compared with other subfamily SNXs, is a conserved, approximately 130 amino acid residue-domain with a specific molecular configuration (Figure 1A). The PXA and PXC domains are largely uncharacterized.

The PX domain of the SNX-PXA-RGS-PXC subfamily is similar to the PX domains of all other SNX subfamilies, with around 100-130 residues, comprised of three β-strands and three α-helices [8]. The conserved sequence ΨPxxPxK (Ψ refers to any large aliphatic amino acid V, I, L, or M) forms a shallow, positively charged proline-rich loop that is considered to be the binding site of the negatively charged phosphate groups of phosphoinositides [4]. Phosphatidylinositol 3-phosphate (PI(3)P), primarily found in early endosome membranes, is a common target of SNXs [9]. This was confirmed from the analysis of the crystal structure of the SNX PX domains [4]. Although PI(3)P is the most common phosphoinositide bound to SNX, many other phosphoinositide interactions have also been demonstrated (Table 2) [5,6,7,8,9,10,11,12,13,14,15,16,17,18,19,20,21,22,23,24,25,26,27]. The PX domain acts not only as a lipid recognition module [7,11], but also plays a key role in protein–protein interactions, such as the interaction of SNX13 and SNX14 with Gαs [4,6] and SNX19 with IA2 [4,6] and D_1_R [20]. However, the molecular details of these interactions remain to be characterized further.

The RGS domain is present in SNX13, SNX14, and SNX25, but not SNX19 [6] (Table 3). This domain is found in a number of molecules, including 20 canonical mammalian RGS proteins and an additional 19 proteins that mediate the interaction with GPCRs or Gα subunits [28]. G proteins are activated by the binding of GTP to Gα and separation from the Gβγ dimer; the deactivation of G proteins occurs when GTP is hydrolyzed by the action of the GTPase-activating proteins (GAPs) (Figure 1B). RGS proteins bind to Gα to facilitate the GTP hydrolysis, accelerating the termination of G protein signaling [29,30]. The SNX-PXA-RGS-PXC subfamily belongs to 19 noncanonical proteins that were previously considered nonfunctional [31]. Recent findings demonstrated that the RGS domain in SNX proteins, like canonical RGS proteins, is involved in the attenuation of GPCR and related G protein signaling [13,32,33]. 

## 3. SNX-PXA-RGS-PXC Subfamily in Receptor Signaling

Similar to the canonical RGS proteins [34], the RGS domain in this subfamily functions as a GAP module, which potentially attenuates GPCR signaling (Table 3 [13,20,23,27,33,35,36]). SNX13 is the first identified SNX that contains the RGS domain, which regulates signaling triggered by GPCRs [33]. Zheng et al. reported that SNX13, through its RGS domain, interacts with the constitutively active form of Gαs, accelerating the hydrolysis of GTP by Gαs [33]. Exogenous expression of the RGS domain of SNX13 reduces the agonist-mediated cAMP increase in HEK293 cells and adenylate cyclase activity in rat cardiac membranes [32,33], while no effect is observed on forskolin-induced cAMP production and adenylate cyclase activity [33], which does not require Gαs. These studies confirm the role of SNX13, as a GAP, in attenuating Gαs-mediated signaling, indicating that SNX13 plays a critical role in the regulation of the duration of GPCR signaling [32]. SNX13 and D_1_R may interact because *SNX13* 105820C and *DRD1* G-94 have been associated with an increase in albumin excretion in a twin pair study [37]. Therefore, SNX 13 may have a role in D_1_R signaling.

The RGS domain does not have to function as a GAP to regulate GPCR signaling in all cases. For example, the RGS domain of SNX14 does not have GAP activity, but specifically binds to and sequesters Gαs, inhibiting the downstream cAMP production caused by the activation of serotonin receptor 6 (5-HT_6_R) [13]. The binding affinity of SNX14 for Gαs is markedly attenuated by the phosphorylation of the RGS domain [13]. This suggests that SNX14 negatively regulates 5-HT_6_R signaling by sequestering Gαs.

As discussed above, the RGS domain facilitates the SNX-PXA-RGS-PXC subfamily in the regulation of GPCR signaling by sequestering Gαs with [33] or without [13] GAP function. To confirm further that RGS domain is not always required for SNX regulation of GPCR signaling, it is critical to study SNX19, a member of this family without RGS domain. SNX19 is essential for the lipid raft residence of D_1_R, cAMP production, and promotion of effective D_1_R signaling [20]. SNX19 also regulates the signaling of histamine receptor H4 (HRH4), a GPCR that is important in the initiation and maintenance of inflammation in mouse lung, following ammonia exposure [38].

In addition to GPCRs, the SNX-PXA-RGS-PXC subfamily also regulates the signaling of non-GPCRs. In mouse insulinoma cells exposed to high glucose concentration, SNX19 inhibits the conversion of PI(4,5)P_2_ to PI(3,4,5)P_3_ and suppresses the phosphorylation of Akt/protein kinase B (PKB), playing critical roles in insulin receptor signaling [22]. In NIH3T3 fibroblasts, SNX25 negatively interacts with transforming growth factor-β receptor 1 (TGF-β1) and downregulates its signaling by increasing the degradation of its receptor [23]. Of note, RGS domain is not necessarily responsible for the regulation of signaling [20,23]. Deletion of either PX or PXA domain abolishes the interaction of SNX25 with TGF-β1 and inhibits TGF-β1 signaling [23]. However, the RGS domain is not critical for the regulation of receptor signaling in this context [23]. SNX25 may also be involved in the circadian rhythmic regulation of vasopressin secretion in the mouse suprachiasmatic nucleus [24].

## 4. SNX-PXA-RGS-PXC Subfamily in Membrane Trafficking

Upon endocytosis, receptors (GPCR or non-GPCR) are trafficked to early endosomes, and then sorted to distinct destinations: lysosomal-mediated degradation or recycling to the plasma membrane or other organelle compartments for reuse [39]. As discussed previously, the SNX-PXA-RGS-PXC subfamily has a conserved PX domain, which enables the SNX to be targeted effectively to endosomal membranes, most frequently by binding to PI(3)P [6]. Therefore, the SNX-PXA-RGS-PXC subfamily represents a core regulator for mediating receptor-endocytic membrane trafficking.

### 4.1. SNX-PXA-RGS-PXC Subfamily in Lysosomal-Mediated Degradation

Endolysosomal trafficking is the major pathway by which transmembrane receptors are downregulated. Membrane contact sites (MCS) between lysosomes and endosomes, as well as mitochondria and endoplasmic reticulum (ER), are regions of phospholipid exchange, which regulate the sorting of receptors at late endosomes for degradation [40,41]. In yeast, Mdm1 (mitochondrial distribution and morphology 1), equivalent to the mammalian SNX-PXA-RGS-PXC subfamily, is a tethering protein that localizes to ER-vacuole/lysosome MCS [42]. Mdm1 PX domain is required and sufficient for its association with the vacuole/lysosome surface [42]. Overexpression of Mdm1 induces ER-vacuole/lysosome tethering and truncation of Mdm1, which removes the PXA domain, disrupts the ER-vacuole tethering, and suppresses lipid exchange and endolysosomal sorting [42].

SNX13 binds to a wide range of phosphoinositides (Table 2) and plays an important role in receptor-endosome-lysosomal degradation. In zebrafish cardiomyocytes, a reduction in SNX13 expression promotes the endolysosomal sorting of apoptosis repressor with caspase recruitment domain (ARC) for its lysosomal degradation [12]. SNX13 interacts with ARC and regulates the interaction between ARC and caspase-8. The increase in the lysosomal degradation of ARC results in the removal of ARC-mediated inhibition and the activation of caspase-8, leading to the activation of the extrinsic apoptotic pathway and subsequent apoptotic cardiomyocyte death [12]. In HEK293 cells, overexpression of SNX13 delays the ligand-dependent EGFR lysosomal targeting, trafficking, and degradation [33], similar to the knockdown of Gαs by RNA interference [35]. SNX13 colocalizes with Gαs and hepatocyte growth factor-regulated tyrosine kinase substrate (Hrs) [35], a critical component of the endosomal sorting machinery for sequestration into multivesicular bodies and subsequent degradation in lysosomes [43]. Henceforth, SNX13 effectively promotes EGFR lysosomal degradation.

Morphological evidence also demonstrated the critical role of SNX13 in lysosomal degradation. Two distinct endosome morphologies, vesicular and tubular, are involved in receptor degradation and recycling pathways, respectively [44]. An unusually abundant amount of tubular endosome structures was observed in the visceral yolk sac endoderm cells of systemic *Snx13*-null mice [45]. This indicates that the receptor is rerouted from endosomes to recycling or TGN pathways due to the defect in the sorting of the lysosomal pathway from early endosomes caused by the knockout of *SNX13*.

The SNX14 PX domain preferentially binds to PI(3,5)P_2_ [16], a key component of late endosomes/lysosomes [9,10], implicating its role in lysosomal degradation [16]. Similar to the yeast homologue of SNX14, Mdm1, which mediates the formation of ER-vacuole contact sites [42], SNX14 tethers for ER localization through its N-terminal transmembrane helices [14]. Knockdown of *SNX14* causes accumulation of aberrant cytoplasmic vacuoles, suggesting defects in endolysosomal homeostasis [14]. SNX14 localizes at the interface between the ER and lipid droplets (LDs); SNX14, overexpressed in human bone osteosarcoma epithelial cells (U2OS), mediates LD budding and growth from the ER surface, after which the LDs are released following its maturation [15]. SNX14 also interacts with 5-HT_6_R, facilitating its endolysosomal degradation [13]. In yeast, Mdm1 not only tethers ER and LDs together, but also generates a high concentration of activated lipids proximal to the vacuole that may facilitate LDs’ autophagic lysosomal degradation [46].

Knockdown of *Snx19* decreases the transmembrane protein, insulinoma-associated protein 2 (IA-2), and the number of dense core vesicles (DCV) in MIN6 cells, a mouse pancreatic β-cell line. The decrease in the IA-2 protein expression and the amount of DCV correlate with the increase in autophagic lysosomal activity [47], which is rescued with the re-introduction of SNX19, indicating a critical role of SNX19 in DCV autophagic lysosomal degradation in MIN6 cells [47].

SNX25 interacts with tropomyosin receptor kinase B (TrkB) in early endosomes, late endosomes, and lysosomes in hippocampal neurons and HEK293T cells [48]. SNX25 overexpression remarkedly reduces the expression of ligand dependent TrkB protein in HEK293T cells [48]. These findings suggest that SNX25 is important in the endolysosomal degradation of TrkB.

### 4.2. SNX-PXA-RGS-PXC Subfamily in Membrane Recycling

Besides mediating endolysosomal degradation, as described above [49], the SNX-PXA-RGS-PXC subfamily, like other SNXs, also regulates receptor membrane recycling. SNX19 plays an important role in D_1_R plasma membrane recycling [20]. In renal proximal tubule cells, SNX19 interacts and colocalizes with D_1_R at the plasma membrane, specifically in lipid rafts. This colocalization is increased by treatment with fenoldopam, a D_1_-like receptor agonist [20]. The increase in their colocalization starts within a few minutes and returns to the basal level after one hour [20]. Depletion of *SNX19* by its specific siRNA decreases D_1_R lipid raft localization, plasma membrane expression, and signaling [20]. All of these results indicate the critical role of SNX19 in D_1_R recycling, probably via palmitoylation and lipid raft targeting.

SNX25 interacts with D_1_R and D_2_R in HEK293 cells, and overexpression of SNX25 perturbs the endocytosis of D_1_R and D_2_R and recycling of the D_2_R. Moreover, knockdown of *SNX25* causes a subsequent decrease in D_2_R plasma membrane expression, suggesting that SNX25 plays a role in D_2_R membrane recycling [27].

## 5. Comparison of SNX-PXA-RGS-PXC Subfamily with SNX-BAR Subfamily in Receptor Signaling and Membrane Trafficking

SNX-BAR, another subfamily of SNXs, is known to regulate receptor signaling and orchestrate membrane trafficking through distinct mechanisms. Although there are few overlaps with the SNX-PXA-RGS-PXC subfamily, the SNX-BAR sorting nexin subfamily regulates different types of receptor cargoes. For example, SNX1 is important for D_5_R signaling [50], while the SNX5 regulates the signaling and trafficking of D_1_R [51], insulin receptors [52], and insulin-degrading enzyme [53] in renal proximal tubule cells. Likewise, SNX1, SNX2, and SNX6 have been found to regulate the membrane trafficking of cation-independent mannose phosphate receptor (CI-MPR) [54,55], cell surface receptor CED-1 [56], TGN38 [57], vacuolar sorting receptor [58], β-site amyloid precursor protein-cleaving enzyme 1 (BACE1) [59], PIN1 [60], and PIN2 [60]. SNX4 regulates the transferrin receptor [61], BACE1 [62], and E-cadherin recycling [63]. SNX18 regulates the transfer of LC3 from the recycling endosome to the autophagosome [64].

Distinct from the SNX-PXA-RGS-PXC subfamily, the SNX-BAR subfamily shares a close relationship with retromers and other retrieval machineries. SNX-BAR subfamily contains a dimeric Bin-Amphiphysin-Rvs (BAR) domain with a positively charged curved surface that binds to membranes [65]. The BAR domain confers targeting to the tubular domain of the endosome, and the endosome aids the transition from a spherical vacuole to a tubule membrane through the interaction of the BAR domains with endosomes, forming a tubular transport carrier [7]. In yeast, the SNX-BAR dimer forms a stable complex with the retromer, a heterotrimer of Vps26-Vps29-Vps35 [65]. In mammalian cells, the association of SNX-BAR dimer with the retromer is relatively weak, but SNX-BAR still relies on the retromer to orchestrate the recognition and capture of specific cargoes [2,3]. The weak association of SNX-BAR with the retromer in mammalian cells may reflect the large diversity of cargoes and the need for other proteins, such as Rab GTPases [66,67], ubiquitin [68], actin filaments [67], and WASH complex [69], to coordinate in the regulation of receptor signaling and trafficking [3,7]. The retromer is also critical for SNX-BAR regulation of receptor endocytic trafficking, retromer-independent receptor plasma membrane recycling, and endosome-to-TGN retrograde trafficking [61,70,71]. Because the SNX-PXA-RGS-PXC subfamily lacks the BAR domain, it does not depend on the retromer or other retrieval machineries to regulate receptor cargo signaling and trafficking.

Different from SNX-BAR, the SNX-PXA-RGS-PXC subfamily (except SNX19) plays some roles similar to RGS proteins. Canonical RGS proteins regulate the signaling of their GPCR cargo, by binding directly to Gαs, and function as a GAP [72]. SNX13, like canonical RGS proteins, can function as a GAP [34], but more studies are needed to determine if this function extends to all members of this subfamily of SNXs, i.e., SNX-PXA-RGS-PXC. As aforementioned, D_1_R signaling is regulated by SNX5, a member of SNX-BAR without the RGS domain. Both SNX5 and SNX19 regulate D_1_R internalization in early endosomes [20,51]. It is unknown whether the two SNXs regulate D_1_R subsequent trafficking and lysosomal degradation. SNX5 and SNX19 differently regulate D_1_R recycling (Figure 2).

SNX5 regulates D_1_R signaling, probably through G protein-coupled receptor kinase (GRK) 4-mediated phosphorylation and desensitization of D_1_R, but not by targeting D_1_R to lipid rafts [51]. As previously stated, SNX19, a member of the SNX-PXA-RGS-PXC subfamily without the RGS domain, is required for the D_1_R-stimulated cAMP production [20]. Therefore, the RGS domain and its GAP function are not essential for the regulation of GPCR signaling by SNX19. SNX19 interacts with the Golgi-associated DHHC-type zinc finger enzyme for D_1_R palmitoylation and targeting into lipid rafts, where adenylate cyclase 6 is located [8], to regulate D_1_R signaling [20]. How SNX5 and SNX19, individually or synergistically regulate D_1_R signaling and internalization and if they regulate the degradation of D_1_R in lysosomes remains to be determined.

In contrast to the SNX-BAR subfamily, the SNX-PXA-RGS-PXC subfamily has a different preference for trafficking routes for its receptor cargoes. Based on recent limited studies, the SNX-PXA-RGS-PXC subfamily mainly transports receptor cargoes, via the endolysosomal pathway for degradation [12,16,35,42,43,45], while the SNX-BAR family mainly retrieves cargoes away from lysosomal degradation, via recycling pathways from the endosome to the plasma membrane, or retrograde pathways from the endosome-to-TGN [2,3,4,5,6,7,8,73,74,75]. The different trafficking pathways could be due to the distinct microdomain localization of retrieval machineries (e.g., retromers) for retrieval of receptor cargoes from ESCRT (endosomal sorting complex required for transport proteins) for degradation, as demonstrated in the *Caenorhabditis elegans* coelomocyte [76,77]. Whether a particular receptor cargo is sorted for recycling or endosomal degradation is governed largely by the SNX associated with retrieval complexes or the ESCRT machinery [2,3,5]. It is plausible for SNX-BAR family to regulate plasma membrane recycling or retrograde trafficking from endosomes to TGN through the retromer-dependent or retromer-independent (e.g.,: ESCPE-1, endosomal SNX-BAR sorting complex for promoting exit-1) protein machineries [69,78]. Ubiquitination [79,80] and palmitoylation [81,82] are important mechanisms for receptor cargo sorting into the ESCRT-mediated degradation. The D_1_R is regulated by ubiquitination [83] and palmitoylation [20], and as aforementioned, SNX5 [51] and SNX19 [20] regulate D_1_R signaling and trafficking. Therefore, it is possible that ubiquitin-tagged or palmitoylated D_1_R is sequestered by different ESCRT-subunits, using distinct mechanisms for its lysosomal degradation [84].

## 6. Comparison of SNX-PXA-RGS-PXC Subfamily with Other SNX Subfamilies

The SNX-PXA-RGS-PXC subfamily has differences from the other SNX subfamilies in its role in receptor signaling and trafficking. For example, SNX3, a member of the SNX-PX subfamily, interacts with the retromer complex to regulate cargoes, such as the divalent metal transporter 1-II (DMT1-II) recycling from the endosome to TGN [85]. SNX17, a member of the SNX-FERM subfamily, interacts via its FERM domain with cargoes, such as integrins, for endosomal recycling to the plasma membrane [86]. During this process, SNX17 is associated with the Commander complex, an assembly comprised of at least fifteen proteins, including the retriever, a retromer-like structure, consisting of three proteins VPS35L, VPS26C, and VPS29 [87]. SNX27, another member of the SNX-FERM subfamily, interacts simultaneously, via its unique PDZ domain, with retromer subunit and cargo receptors, such as the β_2_AR, to regulate their recycling [88].

## 7. SNX-PXA-RGS-PXC Subfamily in Physiology and Pathophysiology

As aforementioned, the SNX-PXA-RGS-PXC subfamily regulates the signaling and trafficking of internalized cargoes, including GPCRs and non-GPCRs, mainly leading them to endolysosomal degradation [10,13,42]. There is a dynamic coordinated interaction among the recycling, retrograde, and degradative pathways, which maintains normal cellular functions [2,3]. However, if the SNX-PXA-RGS-PXC subfamily, like all other SNX subfamilies, is dysfunctional and disabled to transport receptor cargoes to their appropriate cellular destinations, there will be the impairment of the above-mentioned pathways, which will negatively affect cellular functions, causing disorders, such as those listed in Table 4 [17,18,19,20,23,27,33,38,89,90,91,92,93,94,95,96,97,98].

SNX13 forms a heterotrimeric complex with Gαs and Hrs in endosomes, critical in targeting ubiquitinated membrane cargoes, such as EGFR, for sequestration into multivesicular bodies and subsequent degradation in lysosomes [35,42]. Germline deletion of *Snx13* in mice is embryonically lethal, indicating that SNX13-regulated endocytosis dynamics is essential in mouse development [45]. SNX13 plays a crucial role in preserving cardiomyocyte survival by targeting ARC endolysosomal degradation [12]. SNX13 is associated with skin pigmentation variation in humans [89,99], indicating that SNX13 plays a role in melanin cellular transport and trafficking.

SNX14 is important in normal neuronal excitability and synaptic transmission [90]. SNX14, localized in the lysosome [16], functions as a negative regulator of the signaling and trafficking of 5-HT_6_R [13] and probably other receptor cargoes, as well. SNX14 is also localized at the membrane contact site of ER-lipid droplets in yeast, drosophila, and mammals [14,42,91,100], indicating important roles of SNX14 in lipid drop biogenesis and trafficking of lipid transfer proteins [101].

SNX19 interacts with D_1_R and Golgi-associated DHHC-type zinc finger [20], a palmitoyltransferase in Golgi [102] and, as previously stated, facilitates D_1_R palmitoylation, trafficking from anterograde trafficking, and recycling [20]. This promotes the residence of D_1_R in the lipid rafts [20], where other D_1_R signaling complex components are localized, including GRK4, G proteins, adenylyl cyclases, and effector proteins, such as NADPH oxidase, Na^+^-K^+^-ATPase, and Na^+^-H^+^ exchanger (NHE) 3, for appropriate cellular responses and functions [103,104,105,106]. The PX domain of SNX19 is required for D_1_R targeting to lipid rafts because the deletion of the PX domain results in the D_1_R mistargeting to non-lipid rafts [20]. Moreover, *SNX19* knockdown not only decreases the D_1_R-induced increase in cAMP production, but also abrogates the ability of the D_1_R to inhibit renal tubular sodium reabsorption [20]. Importantly, renal *Snx19* knockdown increases the systolic blood pressure of C57BL/6J mice [20], indicating critical roles of SNX19 on the regulation of blood pressure. SNX19 also interacts with Islet antigen-2 [92], a major autoantigen in type 1 diabetes, and is located in dense-core secretory vesicles that regulate insulin secretion [23]. SNX19 may function as a protective factor against cartilage degradation [21]. A single nucleotide polymorphism of *SNX19*, rs2298566, increases the risk of coronary heart disease [18].

SNX25 is involved in the lysosomal degradation of the TGF-β receptor [23] and the development of temporal lobe epilepsy [25]. SNX25 interacts with and accelerates tropomyosin-related kinase B degradation [48]. SNX25 may also be involved in the regulation of genes associated with mesothelioma [93]. *SNX25* is a potential candidate gene for distal hereditary motor neuropathies [94] and a genetic modifier of the age of onset of familial Alzheimer’s disease [26].

## 8. Conclusions and Perspectives

Emerging evidence has demonstrated that the SNX-PXA-RGS-PXC subfamily and their interacting partners are critical regulators for receptor signaling and membrane trafficking. The receptor cargoes can be GPCRs and non-GPCRs through which cells respond to both extracellular and intracellular stimulation. The complex interaction between cellular signaling and endosomal-based membrane trafficking plays an essential role in maintaining cellular homeostasis and versatile functions. SNX13, 14, and 25 have a unique RGS domain, which presumably serves as GAP, attenuating signals associated with GPCR. It is important to examine the molecular mechanisms of GAP both in vitro and in vivo for all three SNXs of the above subfamily. Current evidence suggests that SNX 19 lacks an RGS domain, indicating that it is unable to serve as a GAP. However, SNX19 has emerged to regulate GPCR in other ways, for example, facilitating D_1_R signaling through palmitoylation. Further studies are needed to determine the precise molecular mechanisms by which SNX19 regulates palmitoylation in the Golgi and the plasma membrane.

Different from retromer-dependent SNXs, which retrieve their cargoes through recycling to plasma membrane, TGN or other organelles in retromer-dependent and -independent mechanisms, the SNX-PXA-RGS-PXC subfamily mainly regulates their cargo receptors for endolysosomal degradation. The SNX-PXA-RGS-PXC subfamily regulates receptor recycling for certain cargoes as well, but the molecular switch that controls the different post-endocytic trafficking routes remains to be identified.

While cellular signaling directs the distinct receptor cargo trafficking routes, cargo trafficking actively shapes the cellular signaling response as well, by altering the location and time of specific signaling events. The incomplete understanding of the role that RGS-PXC SNX plays in cell polarity warrants further research. For example, it is important to understand the exact function of the SNX-PXA-RGS-PXC subfamily in the sorting of D_1_R and renal sodium transporters to different cell surface domains. We need to study how such processes can control polarized apical and basolateral locations and cellular function for sodium transport in the renal proximal tubule and other nephron segments. It is expected that the SNX-PXA-RGS-PXC subfamily, as with other SNXs, plays diverse roles on the regulation of the intricately linked signaling and trafficking for precise cellular functional outputs. Studies in appropriate conditional or non-conditional global knockout and transgenic or gene rescue animal models will advance our understanding of the physiological functions in vivo of the SNX-PXA-RGS-PXC subfamily and their associated pathophysiological disorders, which could lead to potential novel therapies targeting this SNX subfamily.

## Figures and Tables

**Figure 1 ijms-22-02319-f001:**
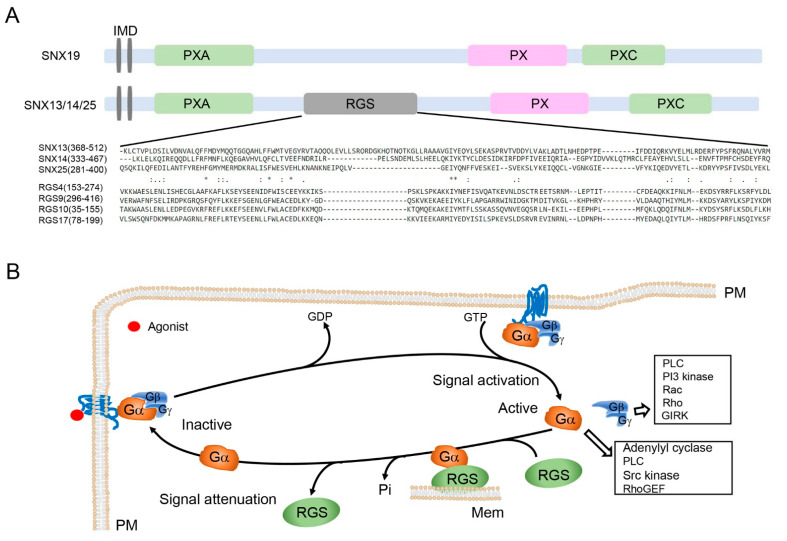
Unique RGS domain in the structure of SNX-PXA-RGS-PXC subfamily. (**A**) Domain organization of SNX-PXA-RGS-PXC subfamily. All members of this subfamily, except SNX19, have unique RGS domains, which are aligned with RGS proteins, as shown. Asterisk denotes identical amino acid residues among all of the seven peptides, one dot indicates the weakly conserved amino acid residues, and double dots indicate the well-conserved amino acid residues among all of the peptides. IMD, integrated transmembrane domain. (**B**) RGS proteins in the G protein nucleotide cycle. Upon agonist binding, receptors activate heterotrimeric G proteins, which induce the exchange of GDP for GTP and dissociation of Gα from Gβγ, attenuation of Gα subunit’s activation (Gα_s_) or inhibition (Gα_i_) of its downstream effectors. The effect is terminated by GTPase or the intrinsic GTPase activity of Gα, where RGS is separated from Gαs. RGS proteins or SNX13 facilitates the hydrolysis of GTP by Gαs, as a GTPase-activating protein. Gα, G alpha subunit; Gβ, G beta subunit; GDP, guanosine diphosphate; Gγ, G gamma subunit; GIRK, G protein-coupled inwardly rectifying potassium; GTP, guanosine triphosphate; Mem, intracellular membranes; PI3 kinase, phosphoinositide 3-kinase; PLC, phospholipase C; PM, plasma membrane; Rac, Rac G protein; RGS, regulators of G protein signaling; Rho, Rho G protein; RhoGEF, guanine nucleotide exchange factor for Rho; Scr kinase, Src family kinases.

**Figure 2 ijms-22-02319-f002:**
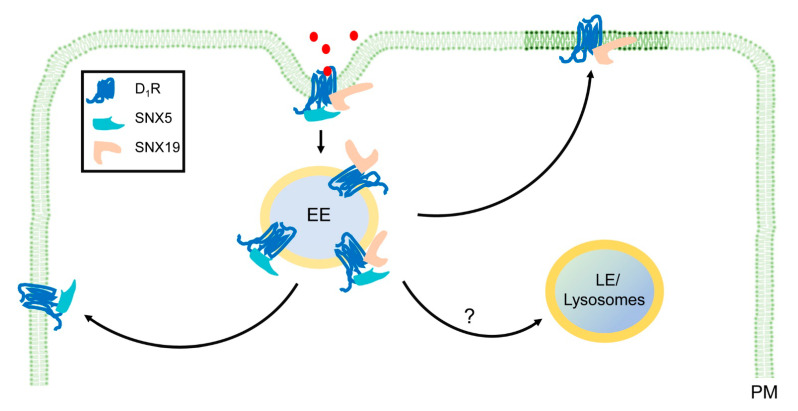
Regulation of D_1_R signaling and membrane trafficking by SNX5 and SNX19 in human renal proximal tubule cells. D_1_R is stimulated by dopamine or D_1_R agonists, resulting in the activation of Gαs and increase in _C_AMP production (not shown). Both SNX5 and SNX19 interact to internalize D_1_R in early endosomes (EE). It is not clear if SNX5 and SNX19 participate in the subsequent trafficking of D_1_R in late endosomes (LE) and lysosomes. SNX5 and SNX19 differently regulate D_1_R recycling; SNX5 may regulate D_1_R recycling through phosphorylation (not shown), while SNX19 regulates D_1_R recycling through palmitoylation and targeting D_1_R to lipid rafts (dark green). Red dots, dopamine, fenoldopam, or other D_1_R agonists.

**Table 1 ijms-22-02319-t001:** Summary of mammalian sorting nexin (SNX) subfamilies [5,6,7,8].

Subfamily *	Members	Major Domain Architecture **	Roles in Signaling, Trafficking, and Degradation
SNX-PXA-RGS-PXC (4)	SNX13, SNX14, SNX19, SNX25	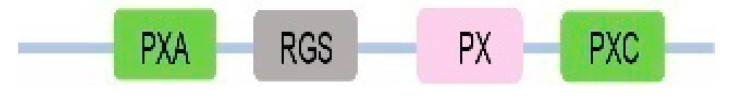	Plays important roles in receptor signaling and membrane trafficking, see text for details.
SNX-FERM (3)	SNX17, SNX27, SNX31	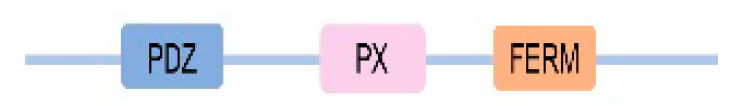	Involved in cargo loading and binding to membrane structures and endosome to plasma membrane trafficking or lysosomal degradation.
SNX-BAR (12)	SNX1, SNX2, SNX4, SNX5, SNX6, SNX7, SNX8, SNX9, SNX18, SNX30, SNX32, SNX33	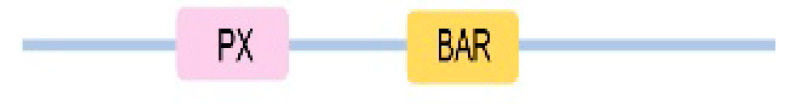	Recognizes and targets to a wide range of cargoes, in coordination with retromers or other retrieval machineries to regulate receptor signaling and trafficking in retromer-dependent and -independent manners.
SNX-PX (10)	SNX3, SNX10, SNX11, SNX12, SNX16, SNX20, SNX21, SNX22, SNX24, SNX29	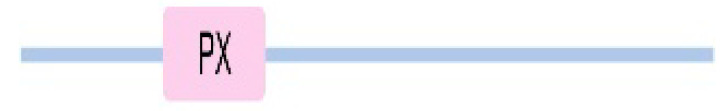	Forms endosome transport carriers in retromer-dependent or -independent manners on a diversity of cargo sorting, retrograde protein trafficking, and lysosomal degradation.
Unclassified SNXs (4)	SNX15, SNX23 SNX26, SNX28	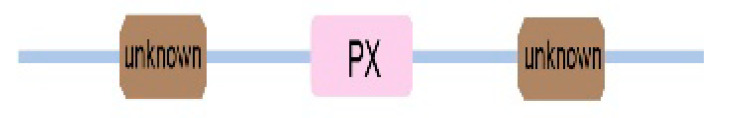	Binds to endosomes in Ca^2+^-dependent or -independent manners; regulates cargoes, such as amyloid-β precursor protein recycling to cell surface and processing for amyloid-β generation.

Note: * The classification is basically dependent on SNX proteins’ domain architecture [6,7,8]. The number in the parenthesis indicates the number of the member proteins of the subfamilies. The unclassified SNX subfamily is comprised of unique SNX members that cannot be conveniently classified into the other four subfamilies. ** For simplicity and clarity, the domain structure is not complete for all subfamily members. For example, the unclassified SNX subfamily has 4 members with structures in addition to PX domain. SNX15 contains a C-terminal MIT domain; SNX23 contains an N-terminal kinesin domain; and SNX26 has a C-terminal SH3 and RhoGAP domain. GAP, GTPase activating protein; MIT, microtubule interacting and trafficking; SH3, Src Homology 3.

**Table 2 ijms-22-02319-t002:** Summary of the characteristics of SNX-PXA-RGS-PXC subfamily members.

SNX	Chromosomal Locus	Major Cellular Distribution	Major Tissue Distribution	Phosphoinositide Binding Preferences	References
SNX13	7p21(human) 12(mouse)	Endosome ER	Pancreas Heart CNS Adipose Spleen	PI(3)P PI(3,4)P_2_ PI(3,5)P_2_ PI(4,5)P_2_ PI(3,4,5)P_3_	[5,6,7,9,10,11,12,13]
SNX14	6q14(human) 9(mouse)	Lysosomes ER Lipid droplets	CNS Adipose Lung Heart Testis	PI(3,5)P_2_	[5,6,7,9,10,11,13,14,15,16,17]
SNX19	11q24.3-q25(human) 9(mouse)	Early endosomes Plasma membrane Mitochondria	Kidney CNS Bone marrow Heart Pancreas	PI(3)P PI(4,5)P_2_ PI(3,4,5)P_3_	[5,6,7,9,10,11,18,19,20,21,22,23]
SNX25	4q35(human) 8A4(mouse)	Endosomes Lysosomes Nucleus	Lung Kidney CNS	PI(3,4)P_2_ PI(3,5)P_2_ PI(4,5)P_2_ PI(3,4,5)P_3_ PI(3)P	[5,6,7,9,10,11,23,24,25,26,27]

Abbreviations: CNS, central nervous system; ER, endoplasmic reticulum; PI, phosphoinositide; SNX, sorting nexin; TGN, trans-Golgi network.

**Table 3 ijms-22-02319-t003:** Examples of RGS domain in SNX-PXA-RGS-PXC subfamily members.

SNX	RGS Domain	Gαs Interaction	GAP Activity	Gαs Signaling	GPCR Cargo Example	Reference(s)
SNX13	+	+	+	inhibition	β_2_-AR EGFR	[33,35]
SNX14	+	+	-	inhibition	5-HT_6_R	[13]
SNX19	-	NA	NA	NA	D_1_R	[20]
SNX25	+	ND	ND	ND	D_1_R D_2_R TGF-β1	[23,27,36]

Abbreviations: SNX, sorting nexin; GAP, GTPase activating protein; GPCR: G protein-coupled receptor; EGFR, epithelial growth factor receptor; AR, adrenergic receptor; 5-HT_6_R, serotonin receptor 6; D_1_R, dopamine receptor 1; D_2_R, dopamine receptor 2; TGF-β1, transforming growth factor β1; +, Yes; -, No; NA, not applicable; ND, not determined.

**Table 4 ijms-22-02319-t004:** SNX-PXA-RGS-PXC subfamily in cellular physiology and implications in diseases.

Subfamily	Signaling	Trafficking Function	Disease Links	References
SNX13	Gαs inhibition	Lysosomal degradation	Saethre-Chotzen syndrome phenotype Type 2 diabetes Skin pigmentation	[33,35,89,95,96]
SNX14	cAMP/PKA inhibition	Lysosomal degradation	SCAR20 Neuron development and differentiation Microcephaly Down syndrome Cerebellar ataxia Intellectual disability Congenital disorders of autophagy Squamous cell carcinoma	[17,90,91,97]
SNX19	Palmitoylation Akt/PKB	Lipid raft targeting Lysosomal degradation Recycling	Hypertension Type I diabetes Atherosclerosis Schizophrenia	[18,19,20,23,92,98]
SNX25	TGFβ-SMAD phosphorylation?	Lysosomal degradation	Temporal lobe epilepsy dHMN LOAD EOAD Hypertension	[25,26,27,94]

Abbreviations: SNX, sorting nexin; Akt/PKB, protein kinase B; cAMP/PKA, cyclic adenosine monophosphate/Protein Kinase A; TGFβ-SMAD, transforming growth factor beta-ALK5-Sma- and Mad-related protein; dHMN, distal hereditary motor neuropathy; EOAD, early-onset Alzheimer’s Disease; LOAD, late-onset Alzheimer’s Disease; SCAR20, Autosomal Recessive Spinocerebellar Ataxia 20.

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
