# Peer review of "SNX-PXA-RGS-PXC Subfamily of SNXs in the Regulation of Receptor-Mediated Signaling and Membrane Trafficking"

_ijms, 2021, doi:10.3390/ijms22052319_

Round 1
Reviewer 1 Report
The review manuscript presented by Bibhas Amatya and colleagues focus on the SNX-PXA-RGS-PXC subfamily of SNXs and their role in signalling and trafficking. The manuscript depicts a vast amount of data available and compartmentalizes the information in several subchapters making the text flow easily. The manuscript is very well written and summarizes the large amount of information available in the subject, which is evident in the number of references provided.
Based on my analysis of the manuscript I recommend it to be accepted for publication with minor corrections that, in my opinion, would greatly improve the quality of the work:
- In my point of view the readers would beneficiate if a list of abbreviations were presented, as several are used along the text;
- Line 36: instead of “in the endosomes”, shouldn’t it be “with the endosomes”?
- Line 66: In the legend of the figure the words “Table 13” appear in the text. I suppose it is an error.
- Line 81: Legend of the table is confusing. What do authors mean with “summary of characterizes of …”? Something must be wrong;
- In table 1, the abbreviation of “CNS” is missing;
- Line 96: In “subfamily is functions” the word “is” should be removed;
- Line 123: Rephrase for clarification such as “participates in the signaling regulation of the ....”
- Line 149: remove the comma between mitochondria and endoplasmic reticulum;
- Lines 161-163: The sentence is rather confusing. The authors should try to rephrase to make it more clear;
- Line 332: the comma is in the wrong place. Must be placed after the word “and” and not before;
- Line 333: Does the authors mean “anterograde” instead of antegrade?
- Last reference lacks numbering.
Reviewer 2 Report
In this manuscript, Amatya et al. review the known roles of the sub-family of sorting nexins (SNXs) termed SNX-PXA-RGS-PXC in the regulation of receptor-mediated signaling and membrane trafficking. SNX-PXA-RGS-PXC forms a subfamily of four proteins, SNX13,14,19,25, which is characterized by the presence, in addition to the PX domain (which binds PIs), of the RGS (regulator of G-protein signaling) domain (except SNX19), and two domains flanking the PX domain, PXA and PXC.
This review depicts in detail the information currently available on this subclass of proteins which are important regulators of receptor signaling (by acting notably on as subunit of G proteins) and trafficking. In particular, the authors insist on the demonstrated role of SNX-PXA-RGS-PXC proteins on receptor trafficking to lysosomes and compare it to the role of another SNX protein family, SNX-BAR, characterized in receptor recycling and retrograde trafficking.
Overall, the review contains a lot of important and up-to-date information. However, it is at times difficult to read because the manuscript is not very well organized. It will benefit from more illustrations and partial reorganization of the manuscript.
Main points:
- It would be good to have a broader view on the SNX protein family with perhaps a table with the main features of each family. An update of Table 1 by Teasdale and Collins (2012) would be welcome. At least the authors should provide the number of members of each family and examples of the best characterized ones.
- What is the role of the N-terminal expected transmembrane domains? One would expect that they play a major role in targeting to organelle membrane. Written line 180 but it should be before. Figure 1 could more explicitly show the relationship between receptor activation and SNX-PXA-RGS-PXC proteins which are also targeted to membranes (and not just free floating RGS). An updated Figure 4 of Teasdale and Collins could serve as a guide.
- A major feature of SNX-PXA-RGS-PXC proteins stressed by the authors is the attenuation of GPCR signaling. It would be important to describe better the experimental evidence for that.
- The authors give a lot of information on the effect of protein KD/KO but it should be more precise to be useful. For example, line 189 “SNX19 plays an important role in D1R plasma membrane recycling”, in which way? It is explained line 208 and up, but should be made clearer from the start. As a matter of fact, D1R could be used as an illustrative example on the role of various SNX proteins, at least SNX13, SNX19 and SNX5.
Parts of the manuscript needing rewriting (non-exhaustive):
PI(4,5)P2 should be rewritten as PI(4,5)P2 and PI3P, as PI(3)P; similarly for the other phosphoinositides.
Please avoid colloquial expressions such as “aka” (line 51)
Line 49; please specify which other SNX subfamilies
Line 52; I would rephrase the sentence “two expected N-terminal helical transmembrane domains” or explain why they are expected.
Line 79; please provide examples of protein-protein interactions
Line 96-97; please check grammar
Paragraph at lines 84-94 should be merged with the one at lines 56-58 as they both talk about RGS domain; as it is now, it is a bit confusing and not linear.
Line 105; please describe how SNX13 regulates signalling of D1R.
Line 175 move the last sentence of the paragraph to line 171.
Lines 202-4 are redundant with preceding text (lines 139-145)
Paragraph at lines 208-214 doesn’t talk about recycling, as one expects from the subtitle at line 201.
Line 225; the differences between SNX-PXA-RGS-PXC and SNX5 in regulating trafficking and signalling of D1R are not described, despite the title of this paragraph.
Line 252-253; redundant, it was already said at lines 118-119.
Lines 257-267 and 284-287 contain some redundant information on D1R and should be put together with what already said about this receptor elsewhere in the manuscript.
Lines 337-338; already stated elsewhere in the manuscript.
In paragraph 6, examples of the roles of other SNX subfamilies members are provided but it is not specified in what sense they differ from SNX-PXA-RGS-PXC.
Line 376,378. “it is vital… It is imperative”. This is quite an overstatement. Again, a full description of D1R trafficking will help understanding the interplay between various SNX proteins.
